# Dyslexia Due to Visual Impairments

**DOI:** 10.3390/biomedicines11092559

**Published:** 2023-09-18

**Authors:** Reinhard Werth

**Affiliations:** Institute for Social Pediatrics and Adolescent Medicine, Ludwig-Maximilians-University of Munich, Haydnstr. 5, D-80336 München, Germany; r.werth@lrz.uni-muenchen.de; Tel.: +49-(0)-1733550232

**Keywords:** dyslexia, children, causes, reading therapy, attention, simultaneous recognition, temporal summation

## Abstract

Reading involves many different abilities that are necessary or sufficient conditions for fluent and flawless reading. The absence of one necessary or of all sufficient conditions is a cause of dyslexia. The present study investigates whether too short fixation times and an impaired ability to recognize a string of letters simultaneously are causes of dyslexia. The frequency and types of reading mistakes were investigated in a tachistoscopic pseudoword experiment with 100 children with dyslexia to test the impact of too short fixation times and the attempts of children with dyslexia to recognize more letters simultaneously than they can when reading pseudowords. The experiment demonstrates that all types of reading mistakes disappear when the fixation time increases and/or the number of letters that the children try to recognize simultaneously is reduced. The results cannot be interpreted as being due to altered visual crowding, impaired attention, or impaired phonological awareness, but can be regarded as an effect of impaired temporal summation and a dysfunction in the ventral stream of the visual system.

## 1. Introduction

Fluent and flawless reading requires the cooperation of various visual abilities that are necessary or sufficient conditions for reading. If a necessary condition or if all sufficient conditions are missing, or if these conditions are not adjusted to each other, fluent and flawless reading is not possible [1]. Some necessary or sufficient conditions for reading are obvious, and it can be concluded from our knowledge of the visual system which conditions are necessary and which are sufficient. In contrast, other necessary or sufficient conditions remain “hidden” and can only be revealed by more sophisticated experiments [1]. The absence of a necessary condition or the absence of all sufficient conditions are causes of reading problems [1]. The most obvious causes of reading problems are refractive errors of the eyes, opacity of the cornea, lens, and vitreous body, retinopathy, diseases of the optic nerve, and homonymous hemianopia [2,3,4,5,6,7,8].

### 1.1. “Hidden” Causes of Dyslexia

Impairments such as eye diseases and postchiasmatic lesions in the visual system resulting in homonymous hemianopia are diagnosed during the clinical routine; however, other possible causes of dyslexia remain hidden. There are many other possible causes of dyslexia that do not become apparent in clinical routine. These include an unusual masking (crowding) effect in the visual field [9,10,11,12,13,14,15,16,17,18,19,20,21], an insufficient ability to expand the visual field of attention [22,23,24,25,26,27,28,29,30,31], an impairment in discriminating auditory stimuli [32,33,34,35,36], and an impaired phonological awareness, which includes different abilities, such as decomposing words into syllables and sounds [37,38,39,40,41,42,43], identifying phonemes in words [44,45,46], naming letters, objects, numbers, and colors [40,47], and rhyming [48]. It has also been hypothesized that dyslexia may be due to a lack of eye movement control during reading [49,50,51,52,53,54,55,56,57,58,59,60,61,62,63,64,65,66,67]. Whether these impairments are causes of dyslexia remains a matter of speculation because these theories are not based on a scientific concept of “causation” that would allow for the experimental criteria that could distinguish between causes of dyslexia and concomitant impairments that have no causal influence on reading performance. When experimental methods are applied that reveal the causes of dyslexia, the above-mentioned hypotheses about the causes of dyslexia are not valid [1,68,69,70]. An experimental method to recognize the causes of a reader’s dyslexia, which can be derived from the definition of the concept of causation [1], is to test under which conditions dyslexic readers can read pseudowords of various lengths correctly [68,69,70] and under which conditions they are unable to do so. It will be shown that the reading performance of children with dyslexia improves to normal when reading conditions are tailored to each child´s impairments, i.e., when fixation times are sufficiently long and the pseudowords do not contain more letters than the reader can recognize simultaneously [68,69,70]. This can be achieved when pseudowords are presented tachistoscopically and the test does not require eye movements.

### 1.2. Objectives

The objectives of the present study are (1) to demonstrate the rate at which a certain kind of reading mistake occurs when the fixation interval is too short and that a prolongation of the fixation interval, as well as reducing the number of letters in words, can normalize the reading performance of dyslexic readers when reading pseudowords. (2) It will be shown that the results of the present study and the results of previous experiments cannot be interpreted as a consequence of impaired visual attention. They support the hypothesis that dyslexia is due to a dysfunction of early processing in neural networks in the visual system, including the visual word form area.

## 2. Frequency and Kind of Reading Mistakes in a Tachistoscopic Pseudoword Test

In the case of a reading impairment due to homonymous hemianopia, the cause of the reading deficiency and the underlying neural lesion can be diagnosed during the clinical routine, while other functional impairments of the visual system resulting in reading impairment can only be brought to light using methods that are uncommon among routine diagnostic methods [1,68,69,70]. An experimental method for recognizing the causes of a reader´s dyslexia that can be derived from the definition of the concept of causation [1] is to test under which conditions readers with dyslexia can read pseudowords of various lengths correctly [68,69,70] and under which conditions they are unable to do so. The following pseudoword test demonstrates that altered temporal summation and impairment in the simultaneous recognition of a sequence of letters that make up a word are causes of dyslexia in children.

### 2.1. Materials and Methods

#### 2.1.1. Children with Dyslexia

One hundred children (71 boys and 29 girls; mean age 119.5 months; SD: 14.9 months) who were diagnosed as dyslexic according to the Zuerich Reading Test [71] participated in the experiments. All children were native German speakers and right-handed. They had no neurological, psychiatric, visual, auditory, or language deficits. The children’s IQs were within the normal range. The children were second- to sixth-graders who knew all the individual letters, had received approximately the same reading instructions, and were expected to read fluently but were far from the required reading ability.

#### 2.1.2. Pseudoword Test

In the present study, data from an earlier study [68] with the same children, in which the kinds of reading mistakes had not yet been evaluated, were reevaluated according to the frequency of different kinds of reading mistakes and the position of the letters in the pseudowords. In this pseudoword test, we used easily pronounceable pseudowords consisting of 3, 4, 5, or 6 letters. The pseudowords consisted of sequences of letters from colloquial German words that are common in second-grade textbooks. Twenty pseudowords of a given length were presented for a given time interval on a computer screen. Pseudowords were used because they can only be read correctly when every letter is correctly recognized, making the number and kind of reading mistakes analyzable. If natural words are used, the reader may only recognize a few letters and guess the remaining ones. Then, s/he may make a correct guess without having recognized all letters. Since the first letter in the pseudowords was not flanked by another letter on the left, and the last letter in the pseudowords was not flanked by another letter on the right, all other letters in the pseudowords were flanked by other letters on both sides (bilateral masking), and the effect of being flanked on one or both sides (visual crowding) was also investigated. 

The presentation times of the pseudowords varied between 250 and 500 ms. The distance between the eyes and the monitor was 40 cm. Each trial began with the presentation of a green fixation mark (luminance: 28 cd/m^2^ on a 68 cd/m^2^ background) at the center of the monitor. The child was instructed to direct his/her gaze towards the fixation mark. Fixation was controlled using an infrared eye-tracking system (IRIS eye tracker; sampling rate: 500 Hz). After a variable time interval between 2 and 4 sec during which the child maintained fixation, the fixation mark disappeared and a pseudoword was displayed so that the middle of the pseudoword was at the same location as the fixation point. The letters were black (luminance: 4 cd/m^2^, background: 68 cd/m^2^, height of letters: 14 mm, space between letters: 4 mm). The children were instructed to read each pseudoword aloud. If the pseudoword was not read correctly, the child was asked to spell and write the pseudoword. The children were instructed not to start pronouncing the pseudoword before they were sure of it. After each pronunciation, the children could correct themselves within 5 to 10 s. Then the next trial began, and the green fixation mark was presented again, followed by the presentation of a new pseudoword of the same length for the same time interval as the previously shown pseudoword. In each experiment, a sequence of 20 pseudowords, consisting of 4 letters each, was presented for 250 ms. If 95% of the pseudowords were recognized correctly, a new sequence of 20 5-letter pseudowords was displayed for 250 ms. If 95% of these pseudowords were pronounced correctly, a new sequence of 20 6-letter pseudowords was displayed for 250 ms. If less than 95% of a sequence of pseudowords was read correctly, a different sequence of pseudowords of the same length was presented; however, the presentation time of each pseudoword was increased by 50 ms. If still less than 95% of this sequence of letters was read correctly, a new sequence of pseudowords of the same length was presented for a time interval that was again 50 ms longer. When less than 95% of a sequence of pseudowords of a given length was read correctly at a fixation interval of 500 ms, a different list of pseudowords was presented, whereby the number of letters in the pseudowords was reduced by one. The fixation times and/or the number of letters to be read were increased or decreased until 95% of the list of pseudowords was correctly recognized. The children´s voices were recorded by a computer to evaluate the number and kind of reading mistakes and the vocal reaction times.

#### 2.1.3. Evaluation and Statistics

The rates of omissions, replacements, or exchanged positions were compared using the Bonferroni–Holm correction of Fischer´s Exact Test. 

### 2.2. Results

The results are summarized in Figure 1A–C, Table 1A–C, and Table 2A–C. The comparisons shown in Figure 1A–C, Table 1A–C and Table 2A–C are based on conditioned probabilities. A given percentage of omitted, replaced, or exchanged letters at a given position in pseudowords of a given length means that the percentage of omitted, replaced, or swapped letters at that position had a given value (%) compared with the percentage of letters at different positions in pseudowords of the same length. The difference in omissions between the letters was *p* ≤ 0.001 for the 4- and 5-letter pseudowords. The difference was also *p* ≤ 0.001 when the first letter was compared to all other letters. Comparisons of omissions, replacements, and exchanged positions in three-letter pseudowords were not significant because there were not enough values. Most of these pseudowords were read correctly. There were no statistical differences in omissions and exchanged positions between the 3rd, 4th, 5th, and 6th letters, or in replacements between the 4th, 5th, and 6th letters. In a previous study [70] with the same children, and in previous studies with different children [71,72], the rate of reading errors decreased when the presentation time was increased. All children were able to recognize at least 95% of the pseudowords when the presentation time was sufficiently prolonged and/or the number of letters in the pseudowords was reduced.

## 3. Discussion

### 3.1. Dyslexia Cannot Be Explained by Visual Crowding, an Impaired Conscious Awareness, or an Impairment in Discriminating Auditory Stimuli

Pseudoword experiments [68,69,70] showed that children with dyslexia are able to recognize 95% of pseudowords when the fixation time is sufficiently increased and when the children do not try to recognize more letters simultaneously than they can. An increase in the number of misread last letters in a word not flanked by letters on both sides compared to misread letters in words flanked by letters on both sides can be expected if letter recognition is impaired by visual crowding [9,10,11,12,13,14,15,16,17,18,19,20,21]. The term “crowding” is used here to refer to a masking effect that impairs recognition of a letter when it is flanked on each side by an adjacent letter [16,21]. In the present study, there was no increase in the rate of omissions, substitutions, or swapping of the positions of the last letters not flanked by letters on both sides (no visual masking) compared to the fourth letter flanked by letters on both sides. If there is no increase in the rate of omissions, substitutions, or swapped positions of the last letters, visual crowding did not affect the misreading of letters in the pseudowords used in the present study.

In a previous study with the same children [68] and in previous studies with different children [69,70], it has been shown that all children with dyslexia were able to read 95% of the pseudowords correctly when the fixation intervals were extended and/or the number of letters that the pseudowords contained was reduced. Increasing fixation intervals and/or decreasing the length of pseudowords are necessary conditions (and therefore causes) for improving reading performance. These studies are based on the definition of the terms “necessary conditions”, “sufficient conditions”, and “cause” [1]. Without these clear scientific concepts, it is impossible to scientifically evaluate which abilities are necessary and which are sufficient for correct reading and which impairments cause dyslexia. An ability is a necessary condition for correct reading if this is only possible in the presence of this ability and if it cannot be replaced by another ability. An ability is sufficient in a reading test when its presence results in correct reading, provided that all the necessary conditions are present and that an ability that is a sufficient condition can be replaced by a different ability that is also sufficient [1]. It must be demonstrated that reading normalizes when all necessary conditions and at least one sufficient condition are present. Recognition of letters flanked on both sides by other letters is a necessary condition for correct reading. Therefore, visual crowding must not be so extreme that letters that are located between two other letters cannot be recognized. Correct reading may be possible despite unusual visual crowding if the fixation time is prolonged and the number of letters is reduced. Then, unusual visual crowding is neither a necessary nor a sufficient condition for incorrect reading, and unusual crowding is not a cause of dyslexia. Zorzi et al. (2012) [15] wanted to show that crowding results in poor reading and that eliminating the crowding effect by increasing the spacing between letters improves reading performance when children read a text. In this study, the location of fixation in the words, the position of the letters on the retina, the number of letters the children tried to read simultaneously, and the fixation times were uncontrolled. It may be that the children read fewer letters at a time and increased the number of eye movements in the reading direction when the spaces between the letters were increased so that the size of the words increased. It has already been shown previously [68,69,70] that reading performance improves significantly when the number of letters a child is trying to recognize simultaneously is reduced. Therefore, the study by Zorzi et al. does not demonstrate that poor reading is caused by a crowding effect. Furthermore, the experiment by Zorzi et al. cannot be compared to the pseudoword test in the present and previous studies, in which the location of fixation in the words, the position of the letters on the retina, the number of letters that the children tried to read simultaneously, and the fixation times were well controlled [68,69,70].

The same is true for the inability to expand the field of attention. It has never been shown that the ability to discriminate auditory stimuli and abilities that are considered to be components of phonological awareness, which include decomposing words into syllables and sounds [37,38,39,40,41,42,43], identifying phonemes in words [44,45,46], naming letters, objects, numbers, and colors [40,47], and rhyming [48], are (1) sufficient conditions for correct reading, and (2) which of these abilities are necessary conditions.

Assumptions that an impairment in the ability to discriminate auditory stimuli or components of conscious awareness are causes of dyslexia are scientifically unfounded according to the scientific definitions of the terms “necessary condition”, “sufficient condition”, and “cause”.

### 3.2. Dyslexia Cannot Be Explained by Impaired Visual Attention

The impairment in the ability to recognize all letters in a pseudoword at one time has also been explained by the assumption of a decrease in visual attention in readers with dyslexia [72,73,74,75,76,77,78,79,80,81,82,83,84,85,86,87,88]. The pseudoword experiment demonstrates that the result cannot be explained by assuming that children with dyslexia focused their visual attention on the first letter in a word and that visual attention decreased from left to right, being the lowest at the right end of the pseudoword [88]. This is contradicted by the finding in many trials that children read the first and last letters of the pseudowords correctly, but repeatedly misread letters in the middle of the pseudowords. This shows that the children were able to extend their field of attention from the first letter to the last letter, but were unable to process the letters in the middle of the pseudowords. Nor can it be assumed that the fixation time was too short to focus attention on the location where the pseudoword appeared. In an earlier study [68] with the same children with dyslexia who read the same pseudowords and in repeated studies with different children with dyslexia [69,70], the fixation times increased from 250 ms to 500 ms. It has been repeatedly demonstrated that with longer fixation times, during which no eye movement occurred, the rate of reading errors decreased so that all children could read at least 95% of the 20 pseudowords correctly, provided that the fixation times were long enough. The children were instructed to focus their attention on the fixation point seconds before the pseudoword appeared, that is, on the location at which they expected the pseudoword to appear. Fixation was controlled from the onset of the fixation.

Even if a string of letters is presented in the foveal and parafoveal regions and the fixation time is sufficiently prolonged, the number of letters that readers with dyslexia can simultaneously recognize is limited. Some subjects were only able to recognize three letters simultaneously, some could only recognize four or five letters simultaneously, and a few children could recognize six letters simultaneously [68,69,70], even if the pseudowords were presented for up to 500 ms. The children also focused their attention between 2 and 4 sec on the location where they expected the pseudoword to appear. Thus, the inability to recognize more than a limited number of letters at a time cannot be attributed to a too short fixation time or an impaired ability to focus their attention on the pseudoword within a given time interval. Since this experiment required no eye movement, the assumption that dyslexia is at least in part due to insufficient eye movement control is not valid.

Extended fixation times of up to 500 ms, during which the children were able to expand their fields of attention or shift their focus of attention, did not sufficiently reduce the rate of reading errors in all children. Only reducing the number of letters that had to be processed simultaneously reduced the rate of all types of errors to the extent that 95% of the 20 words were read correctly. This applies to omitting letters, incorrectly replacing letters with letters that are not present in the pseudoword, changing the position of letters in a word, and increasing the length of a word by adding new letters. These findings are difficult to explain by insufficient shifting of visual attention or unusual parafoveal information processing [68,69,70].

### 3.3. Impaired Temporal Summation Is a Cause of Dyslexia

Taken together, the results of the present experiments support earlier findings [68,69,70,89], showing that the impaired reading capacity of dyslexic readers cannot be explained by unusual visual crowding [9,10,11,12,13,14,15,16,17,18,19,20,21], impaired visual attention [21,22,23,24,25,26,27,28,29,30], an impairment in discriminating auditory stimuli [31,32,33,34,35], or impaired phonological awareness, which includes different abilities, such as decomposing words into syllables and sounds [36,37,38,39,40,41,42], identifying phonemes in words [36,37,38], naming letters, objects, numbers, and colors [40,47], and rhyming [48].

An earlier pseudoword experiment [68] with the same children with dyslexia and pseudoword experiments with different children with dyslexia [69,70] demonstrated that the rate of reading errors decreased when fixation times increased and that all children with dyslexia were able to read at least 95% of the 20 pseudowords correctly when the fixation times were long enough. This means that all kinds of reading mistakes disappeared with sufficiently long fixation times. The finding that dyslexic readers require longer fixation times to complete temporal summation concurs with studies showing that detection and recognition of visual stimuli and visual acuity improve with increasing fixation intervals and increasing temporal summation [90,91,92,93,94,95,96,97,98,99,100,101]. When reading a text, readers with dyslexia terminate fixation too early because they perform premature saccades. When fixating a word or word segment to be read, they make a saccade to the next word or word segment before completing the temporal summation necessary for recognizing the word or word segment being fixated. It has been demonstrated that the execution of premature saccades is not due to an impaired ability to control eye movements. Children with dyslexia can learn to prevent premature eye movements and execute correct reading eye movements within less than 30 min during a computer-based reading therapy session [68,69,70].

Previous pseudoword experiments [68,69,70] showed that impaired temporal summation is a cause of dyslexia. The present experiment also demonstrates what types of reading errors occur and what effect the position of letters in pseudowords has on different types of reading errors when fixation durations are too short and/or when children try to recognize more letters at the same time than they can. Tydgat et al. (2009) [102] investigated serial position functions for the identification of letters in a horizontal array of a quasi-random sequence of 5 consonant letters presented for 100 ms. In this study, the array contained a target letter presented at 1 of the five target positions. Each trial began with the presentation of a mask and fixation bars above and below the mask. The mask disappeared, and a string of five letters was presented, followed by a backward mask. Together with the mask, one letter appeared above the mask and one letter appeared below the mask at one of the possible positions of the array of letters. The subjects were asked to decide which of the two letters was shown in the indicated position by pressing a key. The authors found a W-like serial position function. The position effect found in the present study (Figure 1A–C) is not in agreement with the findings of Tydgat et al. This is not surprising since the methods used in these studies were completely different. Tydgat et al. used an unpronounceable sequence of letters; the array of letters was always presented for only 100 ms; the letters were presented between a forward and a backward mask; eye movements were not recorded so that fixation was not controlled; and the subjects had to choose between two alternative letters presented at a given position. These experimental conditions do not correspond to the experimental conditions of the present study or to the conditions of normal text reading.

The different types of reading errors are the result of insufficient temporal summation in the early stages of visual processing and insufficient processing of an array of letters that a child is trying to read simultaneously in the visual word form area of the fusiform gyrus. While reading, activity is conveyed from the retina to the lateral geniculate nucleus (LGN) and from there to the visual cortical areas. Retinal magnocells project to the two most ventral magnocell layers in the lateral LGN, whereas parvo-retinal ganglion cells send fibers to the other four layers [103,104]. There are so-called “koniocells” in the retina that project to the thin koniocellular layers of the LGN and further on to area V1 of the visual cortex and to the extrastriate middle temporal area (MT/V5) [105,106,107]. Area V1 of the visual cortex projects to V2 and V3, so that the fovea is represented in occipital areas V1, V2, and V3 [108,109].

A fiber projection referred to as the “dorsal stream” consists of fibers that project from area V1 to area V2, from V2 to area V3, and to MT/V5. Fibers from the middle temporal area terminate in the angular gyrus. Area V3 also sends fibers to MT/V5 in the dorsal stream and to V4 in the so-called “ventral stream”. In the ventral stream, V4 projects to the inferior temporal gyrus (IT), which projects to the angular gyrus of the dorsal stream [110]. Temporal summation is predominantly achieved in areas V1, V2, and V3 of the visual cortex and to a lesser extent in areas V4, the anteriorly adjacent area VO, area MT, and the intraparietal sulcus [99,111]. Activity in these areas precedes activity in the fusiform gyrus, superior temporal gyrus, precuneus, and inferior parietal cortex, including the angular gyrus and the inferior frontal gyrus of the left cerebral hemisphere. The influence of presentation times on the activation of areas V1, V2, ventral regions of area V4, and area LO (i.e., an area of the lateral occipital cortex located ventrally and posteriorly to area MT/V5) was demonstrated by Bar et al. (2001) [112]. Activation of areas V1 and V2, ventral regions of area V4, and area LO depended on whether drawings presented between 26 ms and 221 ms were seen vaguely or clearly. Pictures were presented to adult subjects only once or repeatedly. The activation of functional magnetic resonance images (fMRI) of areas V1, V2, ventral regions of area V4, and area LO increased the more clearly the drawings could be seen.

There are many more interconnections in the visual system, so the activity of cells in one visual area may depend on the input from many different areas. Felleman and Van Essen (1992) [113] described 32 areas in the macaque cortex that either contain visually responsive neurons or receive projections from visual areas. There were 305 interconnections between visual areas, 242 of which were bidirectional.

The human visual system, including structures for orthographic processing, has evolved from the visual system of evolutionary ancestors common to monkeys and humans. Therefore, many anatomical and functional features of the visual system, including neural networks for object recognition located in the ventral stream, are similar in humans and monkeys. The similarity between the human and monkey visual systems is also underpinned by the findings of Kato et al. (2014) [114]. They found that there are overlapping expression patterns in human dyslexia-related genes (*ROBO1* and *KIAA0319*, and *CNTNAP2* and *CMIP)* and the expression patterns of these genes in the marmoset brain. Grainger et al. (2012) [115] have shown that monkeys (baboons) can learn to discriminate English words from nonsense combinations of letters. Rajalingham et al. (2020) [116] have demonstrated that there is a population of neurons in the inferotemporal cortex of rhesus monkeys that respond selectively to words and that there are neurons that respond selectively to a nonsense combination of letters. In humans, neural networks that analyze the visual features of stimuli precede those that combine the completed visual analysis of words with sounds. The finding that the ability of dyslexic readers to correctly read pseudowords of a given length depends on fixation time [68,69,70] shows that the processing of pseudowords in the ventral stream requires a longer visual input. The finding that most children with dyslexia recognize only a limited number of letters that make up pseudowords demonstrates that the capacity of ventral stream neurons to process an array of letters simultaneously is impaired in these children. The Visual Word Form Area in the fusiform gyrus, which receives input from the visual cortex, is a pivotal area for simultaneously processing multiple letters. Surgical removal of the visual word form area in the left fusiform cortex resulted in an impaired ability to simultaneously recognize a string of letters [117,118].

Many impairments present in children with dyslexia, their different developmental courses, and the neurobiological networks related to dyslexia have been described [119]. These impairments demonstrate that dyslexia results from a developmental dysfunction of neural networks, including the ventral route of the visual system. The visual system is also interconnected with motor systems to control visually guided actions. Impaired temporal summation precedes executive functions and may contribute to deficits in executive functions found in dyslexia via connections of the visual and motor systems [120].

## 4. Conclusions

Dyslexia can have many causes, some of which are obvious while others are not. Based on a clear concept of “causation”, it is possible to distinguish between causes of dyslexia and concomitant impairments. Several studies [68,69,70] have shown that dyslexia can be caused by a too short fixation time and/or an impairment of simultaneous recognition. The present study shows that letters are omitted, replaced by other letters, or their positions in the word are exchanged when fixation times are too short and/or when readers try to recognize more letters simultaneously than they can. When the fixation times are sufficiently prolonged and/or the number of letters that make up a pseudoword is reduced, dyslexic readers can read 95% of the pseudowords correctly.

## Figures and Tables

**Figure 1 biomedicines-11-02559-f001:**
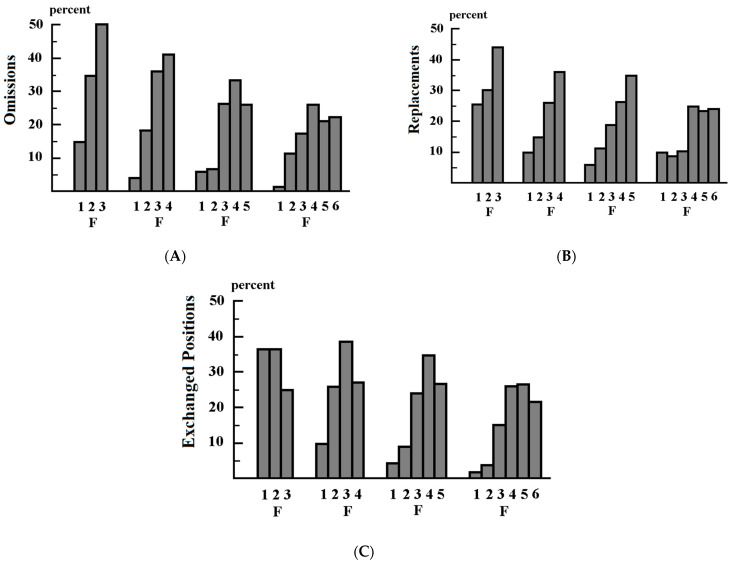
(**A**–**C**): Frequency distribution (%) of the type and position of reading errors in 100 dyslexic children. One hundred percent corresponds to all reading errors that occurred when reading a pseudoword of a given length. The pseudowords had a length of 3, 4, 5, or 6 letters and were displayed between 250 and 500 ms. F: letter that was at the fixation point. 1: first letter at the beginning of the word; 2: second letter from left; 3: third letter from left, etc. (**A**) Frequency of omitted letters; (**B**) frequency of replaced letters; (**C**) frequency of exchanged positions of letters.

**Table 1 biomedicines-11-02559-t001:** (A–C) rate (%) of misread letters in pseudowords. (A) First column: positions of omitted letters in pseudowords. Second to sixth columns: percentage of omitted letters at a given position in pseudowords of a given length. (B) Rate (%) of replaced letters in pseudowords. First column: positions of replaced letters in pseudowords. Second to sixth columns: percentage of replaced letters at a given position in pseudowords of a given length. (C) Frequency distribution of exchanged positions of letters in pseudowords. First row: First column: positions of letters within pseudowords. Second to sixth columns: rate of letters at an exchanged position within pseudowords of a given length.

Position of Letters in the Pseudoword	3-Letter Pseudowords	4-Letter Pseudowords	5-Letter Pseudowords	6-Letter Pseudowords
(A)				
First	15%	3.82%	5.84%	0.88%
Second	34.62%	18.47%	7.29%	12.28%
Third	50%	36.31%	27.00%	17.54%
Forth		41.40%	33.58%	26.32%
Fifth			26.28%	21.05%
Sixth				21.93%
(B)				
First	25.55%	10.06%	6.78%	9.78%
Second	30%	15.24%	12.43%	8.15%
Third	44.45%	27.44%	18.09%	10.33%
Forth		47.26%	27.12%	25.54%
Fifth			35.59%	22.83%
Sixth				23.37%
(C)				
First	37.50%	9.47%	4.55%	2.42%
Second	37.50%	26.32%	9.09%	7.26%
Third	25.00%	38.95%	23.86%	15.32%
Forth		25.26%	35.23%	25.81%
Fifth			27.27%	26.61%
Sixth				22.58%

**Table 2 biomedicines-11-02559-t002:** (A–C) Statistical comparison of the rates of misread letters at different positions in pseudowords. (A) Omitted letters; (B) replaced letters; (C) exchanged positions of letters. Outer left column: position of letters in pseudowords from left to right. Columns 2 to 5: Bonferroni–Holms corrected *p*-values of Fisher´s Exact Test. ns: *p*-values greater than 0.05.

Position of Letters in the Pseudoword	3-Letter Pseudowords Fisher Test Bonferroni–Holms corr.	4-Letter Pseudowords Fisher Test Bonferroni–Holms corr.	5-Letter Pseudowords Fisher Test Bonferroni–Holms corr.	6-Letter Pseudowords Fisher Test Bonferroni–Holms corr.
(A) Omissions				
1st vs. 2nd	ns	*p* ≤ 0.001	ns	*p* ≤ 0.0077
1st vs. 3rd	ns	*p* ≤ 0.001	*p* ≤ 0.001	*p* ≤ 0.0015
1st vs. 4th		*p* ≤ 0.001	*p* ≤ 0.001	*p* ≤ 0.0015
1st vs. 5th			*p* ≤ 0.001	*p* ≤ 0.0015
1st vs. 6th				*p* ≤ 0.0015
2nd vs. 3rd	ns	*p* ≤ 0.001	*p* ≤ 0.001	ns
2nd vs. 4th		*p* ≤ 0.001	*p* ≤ 0.001	ns
2nd vs. 5th			*p* ≤ 0.001	ns
2nd vs. 6th				ns
3rd vs. 4th		*p* ≥ 0.4	ns	ns
3rd vs. 5th			ns	ns
3rd vs. 6th				ns
4th vs. 5th			ns	ns
4th vs. 6th				ns
5th vs. 6th				ns
(B) Replacements				
1st vs. 2nd	ns	ns	ns	ns
1st vs. 3rd	ns		*p* ≤ 0,0095	ns
1st vs. 4th		*p* ≤ 0.001	*p* ≤ 0.001	*p* ≤ 0.0015
1st vs. 5th			*p* ≤ 0.001	*p* ≤ 0.011
1st vs. 6th				*p* ≤ 0.008
2nd vs. 3rd	ns	*p* ≤ 0.001	ns	ns
2nd vs. 4th		*p* ≤ 0.001	*p* ≤ 0.0048	*p* ≤ 0.0015
2nd vs. 5th			*p* ≤ 0.001	*p* ≤ 0.0015
2nd vs. 6th				*p* ≤ 0.0015
3rd vs. 4th		*p* ≤ 0.001	ns	*p* ≤ 0.0022
3rd vs. 5th			*p* ≤ 0.0021	*p* ≤ 0.0133
3rd vs. 6th				*p* ≤ 0.0108
4th vs. 5th			ns	ns
4th vs. 6th				ns
5th vs. 6th				ns
(C) Exchanged Positions				
1st vs. 2nd	ns	ns	ns	ns
1st vs. 3rd	ns	*p* ≤ 0.001	*p* ≤ 0.004	*p* ≤ 0.005
1st vs. 4th		ns	*p* ≤ 0.001	*p* ≤ 0.0015
1st vs. 5th			*p* ≤ 0.001	*p* ≤ 0.0015
1st vs. 6th				*p* ≤ 0.0015
2nd vs. 3rd	ns	ns	ns	ns
2nd vs. 4th		ns	*p* ≤ 0.001	*p* ≤ 0.0015
2nd vs. 5th			*p* ≤ 0.029	*p* ≤ 0.0015
2nd vs. 6th				*p* ≤ 0.0099
3rd vs. 4th		ns	ns	ns
3rd vs. 5th			ns	ns
3rd vs. 6th				ns
4th vs. 5th			ns	ns
4th vs. 6th				ns
5th vs. 6th				ns

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
