# Peer review of "Dyslexia Due to Visual Impairments"

_biomedicines, 2023, doi:10.3390/biomedicines11092559_

Round 1

Reviewer 1 Report

The paper “Dyslexia due to visual impairments” focuses the visual basis of dyslexia. To this aim, the author analyzed neuropsychological data from different tests in children with dyslexia. The results showed how dyslectic performance should be associated with impaired temporal summation, excluding visual crowding, impaired attention, and impaired phonological awareness. The study sounds timely and worth. The results are straightforward. The manuscript offers a general overview of the findings. There might be room for streamline the discussion in order to better sit the study in a broader framework.

1) The introduction risks to be too long and to overload the reader, which might feel lost in too much information. It is suggested to streamline and shrink the introduction in order to more easily guide the reader through the most important concepts up to the logic flow that brings from the need to the rationale of the study.

2) In general, the discussion might be strengthened by providing additional neuropsychological bits to the interpretation of the present study’s results. For example, the author correctly highlights the importance of impaired temporal summation in dyslexia, but a deep/full overview of the anatomo-functional relationship between those two aspects might be better specified. Since the present study is based on visual perception skills, it might be worth mentioning that the obtained results have links to recent discussions about the relationship between brain maturation, visual perception skills, and higher order functions and impairments in humans (Ionta 2021, Frontiers in Human Neuroscience), the involvement of specific brain regions in reading-related skills even in non-human primates (Rajalingham et al 2020 Nat Comm) and  the correspondence of reading-related genes in humans and monkeys (Kato et al 2014 Brain Lang). Providing a similar, more mechanistic, interpretation of how the maturation of the visual system could explain the deficits investigated in the present study, could be beneficial to better understand how the present study sits in a broader framework.

3) The second paragraph of section 4.5 reports the cortical bases of temporal summation, correctly highlighting that the large brain network involved in this function starts from the visual areas and spreads over temporal, parietal, and frontal regions. In this framework, it might be worth noting that hypoactivations of similar large occipito-temporal-parietal networks have been associated with impaired executive functions in people suffering from dyslexia from infancy to adulthood (Farah et al 2021, Frontiers in Psychology). In addition, it has been shown that already in early phases of dyslexia the functional connectivity over large brain networks is impaired (Horowitz-Kraus et al 2018 NeuroImage Clinical). On this basis, it could be proposed that, given the large overlap of the concerned brain networks, impaired temporal summation could at least partially explain the deficits in executive functions found in dyslexia following a bottom-up fashion. In other words, impaired temporal summation might be at the basis of executive dysfunctioning.

Author Response

I like to thank the reviewer for helpful criticism and interesting suggestions.

The paper “Dyslexia due to visual impairments” focuses the visual basis of dyslexia. To this aim, the author analyzed neuropsychological data from different tests in children with dyslexia. The results showed how dyslectic performance should be associated with impaired temporal summation, excluding visual crowding, impaired attention, and impaired phonological awareness. The study sounds timely and worth. The results are straightforward. The manuscript offers a general overview of the findings. There might be room for streamline the discussion in order to better sit the study in a broader framework.

The introduction risks to be too long and to overload the reader, which might feel lost in too much information. It is suggested to streamline and shrink the introduction in order to more easily guide the reader through the most important concepts up to the logic flow that brings from the need to the rationale of the study.

The following lines in the  introduction have been deleted: 40-76, 100-102 and the following lines have been reworded: 6-25, 77-98, 100-102.

The following lines in the discussion have been deleted: 436-451, 535-574, 604-608, 611-612.

The following  lines in the discussion have been reworded: 452-457, 466-469, 482, 488-490, 492-509, 585-600.

The following lines in the „conclusion“ have been reworded: 611-618.

In the section „experiments“, lines 136-219 have been deleted

There is additional text betewwn lines 481 and 482, 575 ff, and between 608 and 611.

Additional text is in red writing

2) In general, the discussion might be strengthened by providing additional neuropsychological bits to the interpretation of the present study’s results. For example, the author correctly highlights the importance of impaired temporal summation in dyslexia, but a deep/full overview of the anatomofunctional relationship between those two aspects might be better specified. Since the present study is based on visual perception skills, it might be worth mentioning that the obtained results have links to recent discussions about the relationship between brain maturation, visual perception skills, and higher order functions and impairments in humans (Ionta 2021, Frontiers in Human Neuroscience), the involvement of specific brain regions in reading-related skills even in non-human primates (Rajalingham et al 2020 Nat Comm) and the correspondence of reading-related genes in humans and monkeys (Kato et al 2014 Brain Lang). Providing a similar, more mechanistic, interpretation of how the maturation of the visual system could explain the deficits investigated in the present study, could be beneficial to better understand how the present study sits in a broader framework.

This has been done in the introduction and in the discussion (lines 84 ff and lines 589 ff.

3) The second paragraph of section 4.5 reports the cortical bases of temporal summation, correctly highlighting that the large brain network involved in this function starts from the visual areas and spreads over temporal, parietal, and frontal regions. In this framework, it might be worth noting that hypoactivations of similar large occipito-temporal-parietal networks have been associated with impaired executive functions in people suffering from dyslexia from infancy to adulthood (Farah et al 2021, Frontiers in Psychology). In addition, it has been shown that already in early phases of dyslexia the functional connectivity over large brain networks is impaired (Horowitz-Kraus et al 2018 NeuroImage Clinical). On this basis, it could be proposed that, given the large overlap of the concerned brain networks, impaired temporal summation could at least partially explain the deficits in executive functions found in dyslexia following a bottom-up fashion. In other words, impaired temporal summation might be at the basis of executive dysfunctioning.

These interesting papers have been included in the last paragraph of the discussion.

The paper of Horowitz-Kraus, T.; Woodburn, M. Rajagopal, A.; Versace, A.; Kovatch, R. et al. Decreased functional connectivity in the fonto-parietal network in children with mood disorders compared to children with dyslexia during rest: an FMRI study. Neuroimage Clinical 2018, 18, 582-590. Doi. org/10.1016/j.nicl.2018.02.034 is in line with many other fmri-studies that I reviewed in an earlier paper (Werth 2021) and which I prefer not to repeat in the present paper.

Reviewer 2 Report

General comment

The paper describes a set of experimental data, some gathered on hemianopic patients and some on dyslexic children.  There are some interesting observations.  However, I personally find the overall layout of the manuscript unclear and confusing.  One has the strong impression that two distant issues (hemianopia and dyslexia) are presented together but the theoretical motivation for this coupling is generally weak and unconvincing. Overall, one has the impression of two distinct presentations which are only juxtaposed.   My personal suggestion is to write separate papers on the different topics. Alternatively, one should find and present a convincing theoretical framework that would allow seeing the merits of a compounded presentation.

Much of the discussion deals with the discussion of different hypotheses of dyslexia (e.g., crowding, impaired visual attention, impaired phonemic awareness, etc.).  However, these comments have very little to do with the actual data presented.  Along the same line, the author endorses the hypothesis of impaired temporal summation; however, the data say very little on this, at least in the way they are currently presented (in particular, no data on presentation times have been reported).

Specific comments

Lines 151-157

The method for this study is presented in a very sketchy fashion; i.e. it would be extremely difficult for another researcher to reliably repeat the same study.  Please provide relevant information on the therapy carried out.

Lines 158-161.  Results are presented in a very sketchy way.   The results of PD are not shown.  Also, in Figure 2 one can appreciate a clear difference in eye movements between before and after therapy.  However, the two conditions shown are also different.  So, it is not immediate for the reader to understand the actual impact of rehabilitation.

Line 195.  The indications relative to the Goldmann perimetry are clearly spelled out for these two patients.  However, note that the same was not done for the first two patients (PD and CG).  Please make the presentation more homogeneous.

Lines 203-208.  A few additional details in the training method would be informative.  For example, what was the task of the subject?  Was there a difference between moving and stationary targets in the results? Etc.

Lines 210-211. “

After 10 days of daily visual field stimulation, the seeing macular region in the lower right quadrand had widened from 2.5 deg. of arc eccentricity to 6 deg. of arc eccentricity in patient NR (Fig. 3 A).”

From the figure, it appears a widening in parafoveal area and a constriction in more peripheral areas.  This is not quoted (and is hard to judge as the vertical axis of the figure is not marked).  Or is this second finding considered a random fluctuation?  Data should be presented more explicitly.

Furthermore, results are presented in a descriptive fashion.  Was any statistical test carried out on the improvement shown by the two patients?  This would allow separating real effects from random fluctuations.

“quadrant” not “quadrand”

Line 214. “However, reading still remained slow and halting in patient ME after the reading training, presumably due to damage to left hemispheric areas involved in reading performance.”

No actual data are presented.

Results of the paper-and-pencil version are presented twice.  The first time on lines 251-257, the second one on lines 294 and following.   This repetition makes the presentation unclear.

Line 305-306.  The legend of the figure is a comment on the results, not a description of the content of the table. 

However, looking at the end of the table one discovers a second legend with much additional information.  Indeed, this legend seems too long (statistical results should be in the text).

Use only one legend (usually on top of the table, not the bottom) and make the presentation more linear.

Line 394 and following (to 418).

Results are presented in an unclear fashion.

Several statistical tests were carried out.  However, only some are presented.  For example, in the case of omissions, tests for three-letter pseudo-words are not presented; tests for four-letter pseudo-words are fully presented; tests for five-letter pseudo-words are partially presented (i.e., comparisons refer only to some letter positions but not others), etc. This type of presentation is very difficult to follow.  As an alternative, the author might include a table that displays all the individual statistical tests carried out.

Looking at Figure 4 A one gets the impression that there are more omissions on short pseudo-words and less on longer pseudo-words.  This seems unlikely.  Possibly, this difference depends upon the use of different presentation times.  However, the impact of different presentation times on the observed performance is not shown; so, it is difficult to appreciate the impact of this (relevant) factor.

Note that in the discussion the role of presentation times is discussed.  For example: “With longer fixation times, the rate of all types of reading errors decreased so that all children could read at least 95% of 20 pseudowords correctly” (lines 492-494).  This makes the presentation very confusing.

Table 2.  In the table legend reference is made to “pseudo-words”.  However, in the table reference is made to “words”.  Please correct this confusion.

Line 470 and following.

This par (“4.2. Dyslexia cannot be explained by visual crowding”) is not easy to understand.  It refers to the lack of effect of letters flanked on the side of the target (indicated as a “word” not a pseudo-word as actually used in the present experiment).   For example: “There was no decrease in the rate of omissions or substitutions of the first or last letter in a word not flanked by letters on both sides….”.

However, there is no mention in the method of the experiment that stimuli were flanked on the side of the targets.

Thus, these comments do not seem to refer to the present study but maybe to a different study by the same author.  Alternatively, the author is commenting on data that were gathered but not presented in the text. Either way, the presentation is very confusing and no conclusion on crowding can be obtained from a study that did not test crowding.

Line 482 and following

This par (“4.3. Dyslexia cannot be explained by impaired visual attention”) suffers from the fact that data on the role of presentation times were not presented in the Results section (see also previous comment).  Thus, it is difficult to appreciate the impact of the present data in evaluating the attentional hypothesis of dyslexia.

Line 575 and following

The par (“4.5. Impaired temporal summation is a cause of dyslexia”) is very hard to follow:

“Taken together, the results of the present experiments support earlier findings [117-119, 151] showing that the impaired reading capacity of dyslexic readers cannot be explained by an unusual masking (crowding) effect in the visual field [56-67], by impaired visual attention [120-136], by an impairment in discriminating auditory stimuli [78-83], or by impaired Phonological Awareness which includes different abilities, such as decomposing words into syllables and sounds [84-90], identifying phonems in words [64-66], naming letters, objects, numbers and colors [94, 95], and rhyming [96]. In patients with a homonymous hemianopia the distortion of visual space has no influence on reading performance [39-55].”

Many of the factors indicated were not tested in the present study, including masking, auditory stimuli, phonemic awareness etc.  The conclusion may well be correct but has a very weak (if any) link to the experiments presented.

phonems should be phonemes.

Line 585.

“The result of the pseudoword experiment demonstrates that impaired temporal summation is a cause for dyslexia. Readers with dyslexia need longer fixation times but perform premature saccades.”

Data on temporal summation were not presented.  Similarly, eye fixation was controlled during the experiment, but actual saccades were not expected and were not reported anyway. 

The conclusion that temporal summation is a cause of dyslexia is not supported by the present data.  Possibly, it is supported by other data, but it seems critical to make clear the contribution of the present data not to report conclusions drawn on other sets of data.

I would suggest the author to carefully check the spelling of the text as I detected several mispellings.

Author Response

I like to thank the reviewer for helpful criticism and suggestions

Comments and Suggestions for Authors General comment The paper describes a set of experimental data, some gathered on hemianopic patients and some on dyslexic children. There are some interesting observations. However, I personally find the overall layout of the manuscript unclear and confusing. One has the strong impression that two distant issues (hemianopia and dyslexia) are presented together but the theoretical motivation for this coupling is generally weak and unconvincing.

Overall, one has the impression of two distinct presentations which are only juxtaposed. My personal suggestion is to write separate papers on the different topics. Alternatively, one should find and present a convincing theoretical framework that would allow seeing the merits of a compounded presentation.

I have deleted a lot of text and I have added new text. New text in red writing.

In detail:

I deleated paragraph 2 in order to make the paper more homogenous.

The following lines in the  introduction have been deleted: 40-76, 100-102 and the following lines have been reworded: 6-25, 77-98, 100-102.

In the section „experiments“, lines 136-219 have been deleted.

The following lines in the discussion have been deleted: 436-451, 535-574, 604-608, 611-612.

The following  lines in the discussion have been reworded: 452-457, 466-469, 482, 488-490, 492-509, 585-600.

The following lines in the „conclusion“ have been reworded: 611-618.

There is additional text beteween lines 481 and 482, 575 ff, between 608 and 611,

Much of the discussion deals with the discussion of different hypotheses of dyslexia (e.g., crowding, impaired visual attention, impaired phonemic awareness, etc.). However, these comments have very little to do with the actual data presented. Along the same line, the author endorses the hypothesis of impaired temporal summation; however, the data say very little on this, at least in the way they are currently presented (in particular, no data on presentation times have been reported).

This has been explained below.

Specific comments
Lines 151-157
The method for this study is presented in a very sketchy fashion; i.e. it would be extremely difficult for another researcher to reliably repeat the same study. Please provide relevant information on the therapy carried out.

This paragraph has been deleted.

Lines 158-161. Results are presented in a very sketchy way. The results of PD are not shown. Also, in Figure 2 one can appreciate a clear in eye movements between before and after therapy. However, the two conditions shown are also different. So, it is not immediate for the reader to understand the actual impact of rehabilitation.

Line 195. The indications relative to the Goldmann perimetry are clearly spelled out for these two patients. However, note that the same was not done for the first two patients (PD and CG). Please make the presentation more homogeneous.
2 Lines 203-208. A few additional details in the training method would be informative. For example, what was the task of the subject? Was there a difference between moving and stationary targets in the results? Etc.

This pragraph has been deleted.

Lines 210-211. “ After 10 days of daily visual field stimulation, the seeing macular region in the lower right quadrand had widened from 2.5 deg. of arc eccentricity to 6 deg. of arc eccentricity in patient NR (Fig. 3 A).” From the figure, it appears a widening in parafoveal area and a constriction in more peripheral areas. This is not quoted (and is hard to judge as the vertical axis of the figure is not marked). Or is this second finding considered a random fluctuation? Data should be presented more explicitly. Furthermore, results are presented in a descriptive fashion. Was any statistical test carried out on the improvement shown by the two patients? This would allow separating real effects from random fluctuations. “quadrant” not “quadrand”

This paragraph has been deleted

Line 214. “However, reading still remained slow and halting in patient ME after the reading training, presumably due to damage to left hemispheric areas involved in reading performance.” No actual data are presented.

This paragraph has been deleted.

 Results of the paper-and-pencil version are presented twice. The first time on lines 251-257, the second one on lines 294 and following. This repetition makes the presentation unclear.

The repitition has been deleted.

Line 305-306. The legend of the figure is a comment on the results, not a description of the content of the table. However, looking at the end of the table one discovers a second legend with much additional information.

Has been corrected

Indeed, this legend seems too long (statistical results should be in the text). Use only one legend (usually on top of the table, not the bottom) and make the presentation more linear.

Has been shifted to „Results“

 Line 394 and following (to 418). Results are presented in an unclear fashion. Several statistical tests were carried out. However, only some are presented. For example, in the case of omissions, tests for three-letter pseudo-words are not presented; tests for four-letter pseudowords are fully presented; tests for five-letter pseudo-words are partially presented (i.e., comparisons refer only to some letter positions but not others), etc. This type of presentation is very difficult to follow. As an alternative, the author might include a table that displays all the individual statistical tests carried out.

I followed the referee´s suggestion and introduced three new tables (Tables 3  A ,B, C) which summarize the results of the statistical tests.

Looking at Figure 4 A one gets the impression that there are more omissions on short pseudo-words and less on longer pseudo-words. This seems unlikely. Possibly, this difference depends upon the use of different presentation times.

The comparisons are based on conditioned probabilities. The high rate of omissions does not mean that there are more omissions  in 3-letter pseudowords than in longer pseudowords. It means that the percentage of omitted letters  on position 3 was high compared to the percentatge of omitted letters on different positions in 3-letter pseudowords. This was the predominant reading error with 3-letter pseudowords. This is also in agreement with my observations in clinical routine. I made this

more clear in addiltional text. The results read now: The results are summarized in Figures 1 A, B, C, in Tables 2 A, B, C, and Tables 3 A, B, C. The comparisons shown in Figures 1 A, B, C and in Tables 2 A, B, C and in Tables 3 A, B, C are based on conditioned probabilities. A given percentage of omitted, replaced or exchanged letters at a given position in pseudowords of a given length means that the percentage of omitted, replaced or swapped letters  at that position had a given value (%) compared to the percentage of letters at different positions in pseudowords of the same length.

However, the impact of different presentation times on the observed performance is not shown; so, it is difficult to appreciate the impact of this (relevant) factor. Note that in the discussion the role of presentation times is discussed. For example: “With longer fixation times, the rate of all types of reading errors decreased so that all children could read at least 95% of 20 pseudowords correctly” (lines 492-494). This makes the presentation very confusing.

These results have been reported in an earlier study in which the same children participated (Werth, Rest Neurol Neurosci, 2018) and in repeated studies with different subjects. I made this clear in additional text:

3.1.2. Pseudoword test
In the present study, data from an earlier study in which the kind of reading mistakes has not yet been evaluated have been reevaluated regarding the frequency of different kinds of reading mistakes.

and in the Discussion

In the present study and in previous studies [90-92], it has been shown that all dyslexic children were able to read 95 % of the psyeudowords correctly when the fixation intervals were extended and/or the number of letters which the pseudowords contained was reduced. Increasing the fixation intervals and/or decreasing the length of the pseudowords were sufficient and necessary conditions (and therefore causes) for the improvement in reading performance. …

In an earlier study [Werth 2018] with the same children and in repeated studies with different children [Werth 2019; 2021], it was shown that with longer fixation times, the rate of all types of reading errors decreased so that all children could read at least 95% of the 20 pseudowords correctly.

Table 2. In the table legend reference is made to “pseudo-words”. However, in the table reference is made to “words”. Please correct this confusion.

Has been corrected

Line 470 and following. This par (“4.2. Dyslexia cannot be explained by visual crowding”) is not easy to understand. It refers to the lack of effect of letters flanked on the side of the target (indicated as a “word” not a pseudoword as actually used in the present experiment). For example: “There was no decrease in the rate of omissions or substitutions of the first or last letter in a word not flanked by letters on both sides….”. However, there is no mention in the method of the experiment that stimuli were flanked on the side of the targets. Thus, these comments do not seem to refer to the present study but maybe to a different study by the same author. Alternatively, the author is commenting on data that were gathered but not presented in the text. Either way, the presentation is very confusing and no conclusion on crowding can be obtained from a study that did not test crowding.

This has been made clear with additional text in “Methods”: Since the first letter in the pseudowords was not flanked by another letter on the left, and the last letter in the pseudowords was not flanked by another letter on the right, all other letters in the pseudowords were flanked by other letters on both sides (bilateral masking), the effect of being flanked on one or both sides (visual crowding) was also investigated. 

Line 482 and following 3 This par (“4.3. Dyslexia cannot be explained by impaired visual attention”) suffers from the fact that data on the role of presentation times were not presented in the Results section (see also previous comment). Thus, it is difficult to appreciate the impact of the present data in evaluating the attentional hypothesis of dyslexia.

To make this issue clear, the following text has been added in“ Methods“:

3.1.2. Pseudoword test

In the present study, data from an earlier study in which the kind of reading mistakes has not yet been evaluated have been reevaluated regarding the frequency of different kinds of reading mistakes.

The following text has been added in „Discussion“:

      This is contradicted by the finding in many trials, that children could read the first and the last letters of the pseudowords correctly, but repeatedly misread letters in the middle of the pseudowords. This shows that the children were able to extend their field of attention from the first letter to the last letter, but they were unable to process the letters in the middle of the pseudowords. Nor can it be assumed that the fixation time was too short to focus attention on the location where the pseudoword appeared. In an earlier study [90] with the same dyslexic children who read the same pseudowords, and in repeated studies with different dyslexic children [91,92], the fixation times increased from 250 milliseconds to 500 milliseconds. It has been repeatly demonstrated that with longer fixation times during which no eye movement occurred, the rate of reading errors decrease so that all children can read at least 95% of the 20 pseudowords correctly, provided that the fixation times are long enough. The children were instructed to focus their attention on the fixation point seconds before the pseudoword appeared, i. e., to the location at which they expected the pseudoword to appear. Fixation was controlled from the onset of the fixation.

 Line 575 and following The par (“4.5. Impaired temporal summation is a cause of dyslexia”) is very hard to follow: “Taken together, the results of the present experiments support earlier findings [117-119, 151] showing that the impaired reading capacity of dyslexic readers cannot be explained by an unusual masking (crowding) effect in the visual field [56-67], by impaired visual attention [120-136], by an impairment in discriminating auditory stimuli [78-83], or by impaired Phonological Awareness which includes different abilities, such as decomposing words into syllables and sounds [84-90], identifying phonems in words [64-66], naming letters, objects, numbers and colors [94, 95], and rhyming [96]. In patients with a homonymous hemianopia the distortion of visual space has no influence on reading performance [39-55].” Many of the factors indicated were not tested in the present study, including masking, auditory stimuli, phonemic awareness etc. The conclusion may well be correct but has a very weak (if any) link to the experiments presented. phonems should be phonemes.

These results have been reported in an earlier study in which the same children participated (Werth, Rest Neurol Neurosci, 2018) and in repeated studies with different subjects. I made this clear in additional text in the discussion. Please see text in red writing above.

To make this issue more clear, the following text has been added in the discussion:

In the present study and in previous studies [90-92], it has been shown that all dyslexic children were able to read 95 % of the psyeudowords correctly when the fixation intervals were extended and/or the number of letters which the pseudowords contained was reduced. Increasing the fixation intervals and/or decreasing the length of the pseudowords were necessary conditions (and therefore causes) for the improvement in reading performance. These studies are based on the definition of the terms „necessary conditions”, “sufficient conditions“ and „cause“[1]. Without these clear scientific concepts, it is impossible to scientifically evaluate which abilities are necessary and which are sufficient for correct reading and which impairments cause dyslexia. An ability is a necessary condition for correct reading if correct reading is only possible in the presence of this ability and if it cannot be replaced by another ability. That an ability is sufficient in a reading test means that the presence of this ability results in correct reading provided that all necessary conditions are present and that an ability that is a sufficient condition can be replaced by a different ability that is also sufficient [1]. It must be demonstrated that reading normalizes when all necessary conditions and at least one sufficient condition are present. A mild masking effect (visual crowding) which is not so extreme that letters which are located between two other letters cannot be recognized is a necessary condition for correct reading only if it is provided that the fixation time is long enough and that the number of letters that make up the word is limited. It may be that correct reading is possible despite an unusual masking effect if the fixation time is prolonged and the number of letters is reduced. Then, usual visual crowding is neither a necessary nor a sufficient condition for correct reading, and unusual crowding is not a cause of dyslexia. The same holds for an inability to expand the field of attention. It has never been shown that an ability to discriminate auditory stimuli, and abilities that are considered to be components of Phonological Awareness which include decomposing words into syllables and sounds [57-63], identifying phonemes in words [64-66], naming letters, objects, numbers and colors [67-68], and rhyming [69] are (1) sufficient conditions for correct reading (2) and which abilities are necessary conditions.

Assumptions that an impairment in the ability to discriminate auditory stimuli or components of conscious awareness are causes of dyslexia are scientifically unfounded according to the scientific definitions of the terms  „necessary condition“, „sufficient condition“ and „cause“.

and later:

An earlier pseudoword experiment [90] with the same dyslexic children [91] and pseudoword experiments with different dyslexic children [90; 92] demonstrated that the rate of reading errors decreased when fixation times increased and that all dyslexic children were able to read at least 95% of the 20 pseudowords correctly when the fixation times were long enough. This means that all kinds of reading mistakes disappeared with sufficiently long fixation times. The finding that dyslexic readers require longer fixation times to complete temporal summation concurs with studies that have shown that detection and recognition of visual stimuli and visual acuity improve with increasing fixation intervals and increasing temporal summation [112-123]. When reading a text, readers with dyslexia terminate fixation too early because they perform premature saccades. When fixating a word or word segment to be read, they make a saccade to the next word or word segment before temporal summation necessary for recognizing the word or word segment being fixated is completed. It has been demonstrated that executing premature saccades is not due to an impaired ability to control eye movements. Dyslexic children can learn to prevent premature eye movements and execute correct reading eye movements within less than 30 min during a computer-based reading therapy session [90-92].

Line 585. “The result of the pseudoword experiment demonstrates that impaired temporal summation is a cause for dyslexia. Readers with dyslexia need longer fixation times but perform premature saccades.” Data on temporal summation were not presented. Similarly, eye fixation was controlled during the experiment, but actual saccades were not expected and were not reported anyway. The conclusion that temporal summation is a cause of dyslexia is not supported by the present data. Possibly, it is supported by other data, but it seems critical to make clear the contribution of the present data not to report conclusions drawn on other sets of data.

This has been made clear with additional text:  

In the present study and in previous studies [90-92], it has been shown that all dyslexic children were able to read 95 % of the psyeudowords correctly when the fixation intervals were extended and/or the number of letters which the pseudowords contained was reduced. Increasing the fixation intervals and/or decreasing the length of the pseudowords were sufficient and necessary conditions (and therefore causes) for the improvement in reading performance. …

and new text above.

Comments on the Quality of English Language I would suggest the author to carefully check the spelling of the text as I detected several mispellings.

Has been corrected

Reviewer 3 Report

This study was conducted systematically to achieve the purpose of the study. Table 1 needs to be modified to increase readability. It would be useful to refer to similar tables reported previously. It would be useful to suggest the limitations of the study.

Author Response

This study was conducted systematically to achieve the purpose of the study. Table 1 needs to be modified to increase readability. It would be useful to refer to similar tables reported previously. It would be useful to suggest the limitations of the study.

I reworded Table 1 somewhat. I constructed Table 1 as best I could. Since the computerized line bisection task has not been described before, I could not refer to previous reports of similar line bisection results.

Discussion, first paragraph. The following text has been added:

The results apply only to the hemianopic children who participated in the present study. It cannot be excluded that there are other hemianopic patients who shift spatial visual coordinates much more and that such a shift influences reading ability.

Round 2

Reviewer 1 Report

Accept

Author Response

Reviewer 1 voted "accept".

Reviewer 2 Report

The revision of the paper improved the readability of the manuscript and clarified several critical points.  At the same time, I feel that there are still several critical points that require attention and that need to be accommodated.  Below I detail these points as they appear in the paper (thus mixing major and minor questions).

Line 17 tachystoscopic should be tachistoscopic (see also lines 88 and 208).

Line 42.  Here and elsewhere, the author introduces an extremely large number of references.  Thus, for the observation that spatial relations are impaired in patients with homonymous hemianopia, there are 17 different references.  It is difficult to carefully check all these materials.  At any rate, even from a cursory analysis, it is apparent that some of these references are closely pertinent and others less so.  For example, in this particular case, reference 17 is about normal subjects, not hemianopic patients.  This is not critical.  However, one wonders whether such a large number of references is useful to the readers of the paper.  Thus, I would suggest the author select the few most informative references.  At any rate, if there is a preference to be exhaustive as now, I would ask the author to carefully check that all quoted references indeed refer to the points made in the text.  I use these lines of text as an example but note that this tendency is present throughout the text.

Line 59: If it turns out that children who are good readers perform as poorly as hemianopic children in line bisection tests this indicates that the results of line bisection tests have no significant influence on reading performance in hemianopic children.

This sentence is clear.  However, I find it less clear the one on line 50 and the following:

If children with dyslexia perform significantly worse in line bisection tests than good readers and if they perform even worse than hemianopic children, it cannot be concluded that the worse performance in a line bisection test is a necessary or sufficient condition for poor reading.

Please reword the sentence.

Line 70: “unusual masking (crowding) effect“

I don't think that equating masking and crowding is correct.  On the contrary several authors emphasize the differences between the two mechanisms (e.g., Pelli, D. G., Palomares, M., & Majaj, N. J. (2004). Crowding is unlike ordinary masking: Distinguishing feature integration from detection. Journal of Vision4(12), 12-12).

Line 92 and the following: I understand what is meant but the sentence is not clear and should be checked for fluency.

Line 97 spacial should be spatial.

Line 116. Zuerich should be Zurich (see also line 223).

Lines 116 and the following (“The visual field…”) should not belong to the “2.1.3. Paper-pencil line bisection test“ section.  They can be moved to the one describing children with hemianopia (“2.1.2. Children with homonymous hemianopia”) or a new sub-section could be made.

Line 114 and following.  Small shifts on the midline of a line are described in hemianopia; much larger shifts are expected if the brain lesion produces hemineglect, indeed a frequent consequence at least in the case of right-sided cerebral lesions.  Can the author exclude that the patients had any sign of neglect?  Was it tested?

Line 135 and the following: Comparing the magnitude of the patients ́ tendencies to shift the middle of the lines in the direction of the hemianopia and the magnitude of deviations of the good readers to the left or to the right of the true midpoint yielded p-values between 0.07 and 0.27 for all lines (Table 1).

I have two comments.

First, is the comparison appropriate?  The data of the two groups clearly indicate different things.  In the hemianopic children, one could posit the influence of the field (hence the scoring depends upon the side of the visual field loss).  In healthy children, a relative score (deviations to the left versus to the right) is used.  How can they be compared?  Indeed, if one looks at the means of these subjects one gets the impression that they do not show any deviation from the midline, at least if performances are averaged algebraically. Thus, most means in column 1 (paper and pencil test) are very close to zero.   This last observation leads me to the second comment.

I wonder whether it would be informative to test statistically whether healthy children show a systematic deviation from the midline.  This can be easily done using students’ ts against the null hypothesis of 0 (zero).  Indeed, an inspection of Table 1 seems to indicate that healthy children do not show any deviation from the midline in the paper and pencil test (means are very close to zero).   In the Computerized line bisection test, there seems to be an effect of the direction of movement of the cursor but if data in the R→ L, and L→ R are averaged, again there seems to be no effect.  This can be checked by statistical testing, as spelled out above.

In the absence of a significant deviation from zero in healthy children, one wonders what the general meaning of the present data is.  This is not to say that they are not interesting.  However, if healthy children simply do not make errors, how this comparison can be informative?  Explaining this point is critical and would make the whole argument more convincing.

Line 230: data from an earlier study.

Please add the reference.

Line 233: The sequences of the letters in the pseudowords also occur in colloquial German words.

This statement seems very generic and would hardly pass any “psycholinguistic” check…. How was this appropriateness of the pseudowords judged?  If bigram frequency was not checked, I guess one could say that they appeared plausible to one (or better more) German native speaker.  Please reword the sentence.

Lines 242-243.  Again, there seems to be confusion between masking and crowding.

Please clarify.

The experiment presented is a re-evaluation of a previous study.  To make sure, are the data presented here (e.g., Figure 1 or Table 2) new or were they at least partially published before?  Please state clearly whether there is some overlap in the data presentation and in case what is (as, if that is the case, this may also call for permission from other publishers).

Line 279 and the following (“Results”)

I find this short paragraph not much informative. Essentially the author refers to the Figure and tables but makes no statement about the actual results obtained.  For example, Table 3 presents several statistical tests, but they are not illustrated in any way in the text. 

I would ask the author to present the findings also in written form and not only through the figure and the tables.  What the experimental data presented tell us?  I understand that comments will be presented in the Discussion section.  However, a detailed presentation of the main findings observed seems in order.  This seems to be an important point to consider in a revision.

Line 319.  The good reading children were younger than the patients and they were already good readers but they were not perfect in their ability to bisect horizontal lines.

Indeed, as stated above, their mean performance was very close to zero.  In the absence of a statistical test, this statement seems unfounded.

Line 321. If they were not good line bisectors, but good readers, it can be concluded that poor line bisection ability has no influence on reading performance.

Same as above.  They actually seemed to be “good line bisectors”!

In general, I find the argument in Section 4.1. unconvincing.  I would ask the author to develop the argument more persuasively.

Line 323.  This is also true for children with dyslexia.

In the paper, there are no data on the line bisection tasks of children with dyslexia.  So, I do not understand this sentence.  Does it refer to other studies?  In that case, please quote. Otherwise, please delete it.

Line 335 and the following. In the present study, there was no decrease in the rate of omissions or substitutions of the first or last letter in a word not flanked by letters on both sides (no visual masking), and there was no increase in the rate of omission or substitution of the letter at the fixation point flanked by letters on both sides (visual masking).

First, as stated above, results were not presented analytically.  Thus, these sentences do not rest on actual statistical tests.  This is a serious drawback of the present manuscript.  One must revert to an inspection of the figure and the tables.  If one does that, the idea that “there was no decrease in the rate of omissions or substitutions of the first or last letter in a word” does not seem entirely correct.  It is very clear from Figure 1, that omissions are always lowest for the first letter and substitutions are the lowest for the first letter for 3- to 5-letter pseudo-words (but not for 6-letter pseudo-words).   As stated by the author, no clear effect on the last letter seems present.  However, this does not authorize mixing the two effects as presently done.  At the very minimum, this proves very confusing.  As stated above, the direction of the effects should be presented in the Results section.  In the discussion, the meaning of the findings can be commented upon.

Second, I also find it unconvincing the sentence “there was no increase in the rate of omission or substitution of the letter at the fixation point flanked by letters on both sides (visual masking).”  Here, the situation seems more complex.  Usually, it is expected that crowding does not occur foveally but only peripherally.  Thus, no crowding effect is expected at fixation (note that this is not a parameter that distinguishes between masking and crowding; so, no different predictions are made in this particular case, although the two phenomena are different in other respects).  So, if one considers for example a pseudoword of 5 letters, one expects better performance for both the first and last letter but also for the third letter.   This W-shaped serial position functions for letter recognition is indeed what was reported by several papers (e.g., Ziegler, J. C., Pech‐Georgel, C., Dufau, S., & Grainger, J. (2010). Rapid processing of letters, digits and symbols: What purely visual‐attentional deficit in developmental dyslexia?. Developmental science, 13(4), F8-F14.; Tydgat, I., & Grainger, J. (2009). Serial position effects in the identification of letters, digits, and symbols. Journal of Experimental Psychology: Human Perception and Performance, 35, 480–498). The W-shaped function does not seem to hold in the present experimental data; however, it would seem important to correctly place the present data within the predictions stemming from the pertinent literature.  The whole argument needs to be presented more clearly.

Lines 345. psyeudowords should be pseudowords.

Line 344 and the following: In the present study and in previous studies [90-92], it has been shown that all dyslexic children were able to read 95 % of the psyeudowords correctly when the fixation intervals were extended and/or the number of letters which the pseudowords contained was reduced.

This finding may well be present in previous research.  However, it does not seem to me that the present study contributes to the finding that “all dyslexic children were able to read 95 % of the pseudowords correctly when the fixation intervals were extended”.  No data along these lines are presented in the Results section.  Indeed, specifying which is the contribution of the present findings, as compared to previous observations of the same author, is quite important in evaluating the relevance of the present paper.  Please clarify this point.

Line 362.  “It may be that correct reading is possible despite an unusual masking effect if the fixation time is prolonged and the number of letters is reduced.” 

The author considers the crowding hypothesis as unable to explain dyslexia.  One may note that various papers report that manipulations that reduce crowding (such as the expansion of the letter size or spacing) also improve reading performance (e.g., Zorzi, M., Barbiero, C., Facoetti, A., Lonciari, I., Carrozzi, M., Montico, M., ... & Ziegler, J. C. (2012). Extra-large letter spacing improves reading in dyslexia. Proceedings of the National Academy of Sciences, 109(28), 11455-11459.).   It would seem important to discuss these papers and why the author dismisses this type of evidence.

Line 372 and the following: “Assumptions that an impairment in the ability to discriminate auditory stimuli or components of conscious awareness are causes of dyslexia are scientifically unfounded according to the scientific definitions of the terms „necessary condition“, „sufficient condition“ and „cause“.

As it turns out, I agree with the ideas of the author of the paper.  However, I wonder whether this statement is appropriate here.  In this study, no manipulation of phonological awareness and the like was made.  Does it make sense to make a conclusion on a parameter that was not tested?  I would ask the author to consider deleting this paragraph.  More generally, it would be important in section 4.2 to let emerge the contribution of the present data over and above general theoretical considerations.

Line 376.  4.3. Dyslexia cannot be explained by impaired visual attention.

I understand the general argument made by the author.  Apart from other considerations, I would like to point out that the presentation would be much clearer if the predictions of an attentional interpretation were made in the Introduction.  As they are presented now, they appear as a post-hoc interpretation of the data.

Line 382.  “This is contradicted by the finding in many trials, that children could read the first and the last letters of the pseudowords correctly, but repeatedly misread letters in the middle of the pseudowords.”

This statement seems inconsistent with what is stated in lines 338-339.

Line 390. Repeatly should be repeatedly.

Line 443.  The result of the pseudoword experiment demonstrates that impaired temporal summation is a cause for dyslexia.

I find the hypothesis put forward by the author interesting.  However, as also stated above, the point here is trying to make clear what is the contribution of the “present” data over and above previous papers. 

A general interpretation that goes beyond the present experimental data is certainly interesting and can be included in the Discussion if it is made clear what is proven by the present data and what is a theoretical statement spanning across several experiments.

As far as I can tell, the linguistic presentation can be improved. 

I indicated some misspellings in my review.

Author Response

The English has been checked independently by two native speakers.

Reviewer 3 Report

Thank you for your response. 

Author Response

Reviewer 3 has no further criticisms

Round 3

Reviewer 1 Report

accept

Author Response

Reviewer 1: accept

Reviewer 2 Report

The revision certainly improved the quality of the paper and made it more readable.  In particular, I appreciate that reference to previous empirical and theoretical work by the author has been presented more explicitly and extensively.
I also would like to thank the author for the detailed responses on several points which are critical. At the same time, I should frankly say that I am not convinced by some of the arguments raised.  For example, I am still dubious that the paper-and-pencil experiment is informative of dyslexia given the evident lack of any type of asymmetry in typical developing children.  At any rate, on these aspects, I trust the judgment of the Editor of the journal.

I only notice here that some of the arguments developed by the author in his response letter were not included in the text. Based on various references and personal experience, the author underscores that crowding is some kind of masking effect and this justifies using the expression “unusual masking (crowding) effect“.  If the author considers crowding as part of masking (based on the evidence he quotes in his accompanying letter), I feel it is important that readers are aware of these theoretical bases.  How else are they going to understand such a statement?  Put in other words, I think the author should explain in the text why he refers to masking and crowding the way he does.

Please note that much the same can be said of other arguments in the accompanying letter, such as the comparison between the present experiment and that of Tydgat et al., 2009. If the author believes that the differences in results can be understood based on the somewhat different paradigms, I feel this should be explained in the text.  After all, at several points in his letter, the author emphasizes the importance of replication studies in psychology: "The more experiments are repeated the more reliable are the results. Therefore, it makes good sense to refer to the literature which reports the same or similar experiments."  However, I notice a tendency not to discuss papers which do not report the same results as the ones presented here.   Apart from the Tydgat et al.' (2009) study, much the same can be said of the Zorzi et al. (2012) study.  Again, this study is relevant to the one presented here and this is discussed in the letter but not in the text.  The author concludes that "In the study of Zorzi et al, all this is not controlled. It may e. g. be that children try to read less letters at a time when there are larger spaces between them. This may already improve reading performance. An interpretation in terms of crowding is very questionable."   If that is so, the readers of the paper will only know if these comments will be presented in the manuscript.

The quality of English has been improved in the present revision.

Author Response

I wish to thank the referee for helpful criticism and suggestions.

The revision certainly improved the quality of the paper and made it more readable.  In particular, I appreciate that reference to previous empirical and theoretical work by the author has been presented more explicitly and extensively.
I also would like to thank the author for the detailed responses on several points which are critical. At the same time, I should frankly say that I am not convinced by some of the arguments raised.  For example, I am still dubious that the paper-and-pencil experiment is informative of dyslexia given the evident lack of any type of asymmetry in typical developing children.  At any rate, on these aspects, I trust the judgment of the Editor of the journal.

I also found that the line bisection experiments do not fit well in the paper. Therefore, I have deleted the line bisection paragraphs.

I only notice here that some of the arguments developed by the author in his response letter were not included in the text. Based on various references and personal experience, the author underscores that crowding is some kind of masking effect and this justifies using the expression “unusual masking (crowding) effect“.  If the author considers crowding as part of masking (based on the evidence he quotes in his accompanying letter), I feel it is important that readers are aware of these theoretical bases.  How else are they going to understand such a statement?  Put in other words, I think the author should explain in the text why he refers to masking and crowding the way he does.

The following sentence has been added on 3.1. (Dyslexia cannot be explained…), lines 6-8 of first paragraph: The term “crowding” is used here to refer to a masking effect that impairs recognition of a letter when it is flanked on each side by an adjacent letter [21] (Pelli, D. G., Palomares, M., & Majaj,N. J. (2004). Crowding is unlike ordinary masking: Distinguishing feature integration from detection. Journal of Vision, 4(12), 12-12).

Please note that much the same can be said of other arguments in the accompanying letter, such as the comparison between the present experiment and that of Tydgat et al., 2009. If the author believes that the differences in results can be understood based on the somewhat different paradigms, I feel this should be explained in the text.  After all, at several points in his letter, the author emphasizes the importance of replication studies in psychology: "The more experiments are repeated the more reliable are the results. Therefore, it makes good sense to refer to the literature which reports the same or similar experiments."  However, I notice a tendency not to discuss papers which do not report the same results as the ones presented here.   Apart from the Tydgat et al.' (2009) study, much the same can be said of the Zorzi et al. (2012) study.  Again, this study is relevant to the one presented here and this is discussed in the letter but not in the text.  The author concludes that "In the study of Zorzi et al, all this is not controlled. It may e. g. be that children try to read less letters at a time when there are larger spaces between them. This may already improve reading performance. An interpretation in terms of crowding is very questionable."   If that is so, the readers of the paper will only know if these comments will be presented in the manuscript.

The following text has been added on page 11, 3.3. ,section 3:

Tydgat et al. 2009 investigated serial position functions for the identification of letters in a horizontal array of a quasi-random sequence of 5 consonant letters presented for 100 ms. The array contained a target letter presented at one of the 5 target positions. Each trial began with the presentation of a mask and fixation bars above and below the mask. The mask disappeared and a string of 5 letters was presented followed by a backward mask. Together with the mask, one letter appeared above the mask and one letter appeared below the mask at one of the possible positions of the array of letters. The subjects were asked to decide which of the two letters was shown in the indicated position by pressing a key. The authors found a W-like serial position function. The position effect found in the present study (Figs. 1 A, B, C) is not in agreement with to the findings of Tydgat et al. This is not surprising, since the methods used in these studies were completely different. Tydgat et al. used an unpronunceable sequence of letters, the array of letters was always presented for only 100 ms, the letters were presented between a forward  and a backward mask, eye movements were not recorded so that fixation was not controlled,  the subjects had to choose between two alternative letters presented at a given position. These experimental conditions do not correspond to the experimental conditions of the present study and to the conditions of normal text reading.Each trial began with two vertical

fixation bars, placed above and below the center of a forward

mask. The forward mask consisted of five hash marks and stayed

on the screen for 515 ms. Then the fixation bars and the mask

disappeared, and the array of five characters immediately appeared

for a duration of 100 ms. This was followed by a backward mask,

which was identical to the forward mask and was accompanied by

two characters, one above the mask and one below at one of the

Tydgat et al. 2009 investigated serial position functions for the identification of letters in a horizontal array of a quasi-random sequence of 5 consonant letters presented for 100 ms. The array contained a target letter presented at one of the 5 target positions. Each trial began with the presentation of a mask and fixation bars above and below the mask. The mask disappeared and a string of 5 letters was presented followed by a backward mask. Together with the mask, one letter appeared above the mask and one letter appeared below the mask at one of the possible positions of the array of letters. The subjects were asked to decide which of the two letters was shown in the indicated position by pressing a key. The authors found a W-like serial position function. The position effect found in the present study (Figs. 1 A, B, C) is not in agreement with to the findings of Tydgat et al. This is not surprising, since the methods used in these studies were completely different. Tydgat et al. used an unpronunceable sequence of letters, the array of letters was always presented for only 100 ms, the letters were presented between a forward  and a backward mask, eye movements were not recorded so that fixation was not controlled,  the subjects had to choose between two alternative letters presented at a given position. These experimental conditions do not correspond to the experimental conditions of the present study and to the conditions of normal text reading.

The following text has been added on page 9, end of section 2:

Zorzi et al. (2012) [15]wanted to show that that crowding results in poor reading and that eliminating the crowding effect by increasing the spacing between letters improves reading performance when children read a text. In this study, the location of fixation in the words, the position of the letters on the retina, the number of letters the children tried to read simultaneously, and the fixation times were uncontrolled. It may be that the children read fewer letters at a time and increased the number of eye movements in the reading direction when the spaces between the letters were increased so that the size of the words increased. It has already been shown previously [70-72] that reading performance improves significantly when the number of letters a child is trying to recognize simultaneously is reduced. Therefore, the study by Zorzi et al.  does not demonstrate that poor reading is caused by a crowding effect. Furthermore, the experiment by Zorzi et al. cannot be compared to the pseudoword test in the present and previous studies in which the location of fixation in the words, the position of the letters on the retina, the number of letters which the children tried to read simultaneously, and the fixation times were well controlled [70-72].

Round 4

Reviewer 2 Report

I appreciate that the author of the manuscript has decided to reduce the text omitting the part on line bisection.  The resulting paper is more compact and reads well.  Therefore, I personally believe that it can be accepted for publication as it stands. 

Below I list a few minor residual changes.

Line 18: "Line bisection" can probably now be deleted.

Line 70: tachystoscopic should be tachistoscopic

Line 96: conisted should be consisted

As far as I can tell, the quality of English is now sufficiently good.

Author Response

Below I list a few minor residual changes.

Line 18: "Line bisection" can probably now be deleted.

Has been corrected

Line 70: tachystoscopic should be tachistoscopic

Has been corrected

Line 96: conisted should be consisted

Has been corrected